# The Potential Role of SP-G as Surface Tension Regulator in Tear Film: From Molecular Simulations to Experimental Observations

**DOI:** 10.3390/ijms23105783

**Published:** 2022-05-21

**Authors:** Martin Schicht, Kamila Riedlová, Mercedes Kukulka, Wenyue Li, Aurelius Scheer, Fabian Garreis, Christina Jacobi, Friedrich Paulsen, Lukasz Cwiklik, Lars Bräuer

**Affiliations:** 1Institute of Functional and Clinical Anatomy, Friedrich-Alexander-Universität Erlangen- Nuremberg, Universitätsstr. 19, 91054 Erlangen, Germany; wenyue.li@fau.de (W.L.); aurelius.scheer@fau.de (A.S.); fabian.garreis@fau.de (F.G.); friedrich.paulsen@fau.de (F.P.); lars.braeuer@fau.de (L.B.); 2J. Heyrovský Institute of Physical Chemistry, Czech Academy of Sciences, 182 00 Prague, Czech Republic; kamila.riedlova@jh-inst.cas.cz (K.R.); mercedes.kukulka@uj.edu.pl (M.K.); lukasz.cwiklik@jh-inst.cas.cz (L.C.); 3Department of Physical and Macromolecular Chemistry, Faculty of Science, Charles University, 128 40 Prague, Czech Republic; 4Faculty of Chemistry, Jagiellonian University, 30387 Krakow, Poland; 5Ophthalmology Private Practice, 17, 81034 Nuremberg, Germany; jacobi2001@gmx.de

**Keywords:** tear film, surfactant protein, surface tension, ocular surface, dry eye

## Abstract

The ocular surface is in constant interaction with the environment and with numerous pathogens. Therefore, complex mechanisms such as a stable tear film and local immune defense mechanisms are required to protect the eye. This study describes the detection, characterization, and putative role of surfactant protein G (SP-G/SFTA2) with respect to wound healing and surface activity. Bioinformatic, biochemical, and immunological methods were combined to elucidate the role of SP-G in tear film. The results show the presence of SP-G in ocular surface tissues and tear film (TF). Increased expression of SP-G was demonstrated in TF of patients with dry eye disease (DED). Addition of recombinant SP-G in combination with lipids led to an accelerated wound healing of human corneal cells as well as to a reduction of TF surface tension. Molecular modeling of TF suggest that SP-G may regulate tear film surface tension and improve its stability through specific interactions with lipids components of the tear film. In conclusion, SP-G is an ocular surface protein with putative wound healing properties that can also reduce the surface tension of the tear film.

## 1. Introduction

Dry eye disease (DED, keratoconjunctivitis sicca) is a global problem, afflicting 15% to 20% of the human population in Central Europe, at least 344 million people worldwide, and is one of the most frequent causes of patient visits to eye care practitioners [1,2,3]. Exogenous influences (e.g., contact lenses, air, drugs, and diet) and endogenous factors (e.g., age, hormonal imbalances, and diabetes mellitus) are the main causes of this disease. DED is based upon a discontinuity of the normal tear film [3,4] characterized by cardinal symptoms such as burning, itchy and watery eyes, painful eye pressure, oppressive feelings, foreign body sensations, and light sensitivity. In most cases, DED is the result of a diminished outer lipid layer at the surface of the tear film [5], a condition called evaporative DED. A quantitative deficit of tears, on the other hand, is considered to be somewhat less common. Meibomian gland dysfunction (MGD) is the leading cause of evaporative DED, which affects countless people worldwide (e.g., more than 40 million in the United States alone) and is one of the most common causes of patient visits in ophthalmology [6].

The meibomian glands (MGs) are important in preventing evaporation and premature disintegration of the tear film. The secretory lipid component of the tear film is mainly derived from the MGs and forms a superficial layer under physiological conditions [5,7,8]. MG lipids and proteins functionally serve to form a clear optical surface, are antimicrobial (act as an effective barrier against noxae and colonization by microorganisms), stabilize the tear film, and reduce its evaporation [7,9]. Some of the MG lipids reduce the surface tension of the tear film, thus ensuring a stable optical interface [8].

During the last years, it has been shown that the surfactant proteins (SP)-A, -B, -C, and -D, which are well known from the lung studies, regulate surface activity in the alveoli and have immunological properties with respect to both innate and adaptive immunity, are not only present in the lung but also play a role at many sites in the human body [10,11,12,13,14], including the ocular surface [15,16,17]. In addition to the well-known surfactant proteins (SP)-A, -B, -C, and -D, two additional surfactant proteins have recently been described by us: SP-G and SP-H. They are also produced by tissues of the ocular surface, among others [13,18]. However, little is known about the functions of SP-G in particular.

The first publication by Mittal et al. shows that SP-G is a secretory protein of lung cells, and their preliminary data raise speculation about its regulatory role in the immune response [19]. SP-G is located on human chromosome 6, a susceptibility locus for diffuse panbronchiolitis [20]. The sequence length suggests that SP-G belongs to the SP group of small and hydrophobic proteins (SP-B, SP-C, and SP-H), but it does not share domains with members of this group and the sequence identities are very low (about 10%) [21]. In earlier studies, we have been able to demonstrate by means of molecular dynamics simulations that SP-G can interact with a lipid system after post-translational modifications due to its hydrophobic properties [21]. SP-G has also been shown to be similar to SP-B and SP-C in its physicochemical properties [13].

The aim of the present studies was to find out whether SP-G is a regular component of the human ocular surface, the lacrimal apparatus, MGs and tears in particular, and whether it plays a role in the context of dry eye disease (DED). In addition to molecular biology and immunohistochemistry studies, we performed molecular dynamics simulations of the tear film to investigate the interaction of SP-G with the lipid component of tears and to uncover a possible mechanism of tear film stabilization at the molecular level. In doing so, the insights gained during the modeling and simulation process support and enhance our cell culture experimental studies.

## 2. Results

### 2.1. Detection of SP-G in Tissue of the Ocular Surface, Lacrimal Apparatus and in Tears

Tissues of ocular surface and lacrimal system (lacrimal gland, eyelid, conjunctiva, and cornea) as well as cells (human corneal epithelial cells (HCE) and human conjunctival epithelial cells (HCjE)) revealed amplification of SP-G mRNA (Figure 1A), which was in accordance with Genebank (https://www.ncbi.nlm.nih.gov/genbank/ (accessed on 17 May 2022), NM_205854). Western blot analysis with proteins derived from different tear fluid samples showed distinct and SP-G specific protein bands at 15 kDa, Bronchoalveolar lavage served as a positive control and revealed the already described bands at 15 and 28 kDa (Figure 1B).

### 2.2. Localization of SP-G within the Tissue of the Ocular Surface and in Cultured HCE and HCjE Cells

SP-G was detected in lacrimal- and meibomian gland, corneal, and conjunctival epithelium as well as in the eyelid (Figure 2) by fluorescence (green, left column) and aminoethylcarbazole (AEC) substrate staining (red, right column). Green and red staining represented a positive antibody response of SP-G in all samples. Paraffin-embedded tissue sections of the ocular system (lacrimal gland, eyelid, conjunctiva, and cornea), 5–7 μm in size, were analyzed. Only samples incubated with primary and secondary antibodies only were negative and showed no staining. The insets in the figure show higher magnification of the tissue used. In the lacrimal gland, cells of the tubular system and, to a minor extent, the acinar cells, showed the presence of SP-G (Figure 2(Aa,Ab)). The lining epithelium of the eyelid showed visible strong SP-G reactivity in the basal epithelial cell layer and faint reactivity within the other layers of the epidermis; strong reactivity was visible on the apical surface of the lining cells (Figure 2(Ac,Ad)). In the conjunctiva, epithelial cells of the healthy conjunctiva showed weak reaction (asterisks) with the SP-G antibody (Figure 2(Ae,Af)). In the meibomian glands, meibocytes were SP-G positive, as well as the lining cells of the excretory duct system. In particular, meibocytes at different stages of maturation showed marked reactivity (Figure 2(Ag,Ah)). In the cornea, SP-G reactivity could be detected in the cytoplasm of basal and apical epithelial cells (Figure 2(Ai,Aj)). Endothelial cells also showed marked reactivity against the SP-G antibody (Figure 2(Ak,Al)). Immunocytochemical studies of corneal (HCE) and conjunctival (HCjE) epithelial cells demonstrated that SP-G is distributed in the cytoplasm and perinuclear region, especially as granula (Figure 2B).

### 2.3. Quantification of SP-G in Tears by ELISA and Determination of Tear Surface Tension as a Function of SP-G Protein Concentration

Comparative quantification of SP-G content in tears by Enzyme-Linked Immunosorbent Assay (ELISA) revealed an average value of 0.6411 ng/mg (*n* = 40) in samples from DED patients (hyper-evaporative form, EDE) (Figure 3A). Healthy donors had an average SP-G protein concentration of 0.1529 ng/mg (*n* = 20). Thus, the level of SP-G in TF was significantly increased in DED compared to healthy subjects (Figure 3A). Mean SP-G concentrations in ocular tissue samples quantified by ELISA were 0.991 ng/mg in the lacrimal gland (*n* = 6), 0.826 ng/mg in the conjunctiva (*n* = 5), and 3.578 ng/mg in the cornea (*n* = 4) (Figure 3B).

### 2.4. Putative Surface Activity of SP-G in Tears

SP-G is characterized by surface activity and a certain degree of hydrophobic surface exposure. The putative influence of recombinant SP-G on tear film surface tension was investigated using a spinning drop tensiometer (KRÜSS GmbH, Hamburg, Germany). Changing the concentration of SP-G in the tear film alone resulted in a slight decrease in tear-film IFT from 54 to 49 mN/m (*n* = 3). A concentration of 100, 500, and 1000 ng/mL SP-G in combination with phospholipid solution (Tears Again (TA)) resulted in a significant decrease in tear-film IFT from 37 to 32 mN/m (*n* = 3). The other two samples used as loading controls showed no effect (Figure 3C). In combination with recombinant SP-G (rhuSP-G) in a tear substitute solution (TA), the surface tension is reduced even more than with rhuSP-G alone.

### 2.5. In Vitro Wound Healing Scratch Assay

The cell culture-based wound healing scratch test (*n* = 4) mimicked corneal epithelial cell damage and demonstrated the effect of rhuSP-G on wound closure rate in combination with phospholipid solution (Tears Again (TA)) (Figure 4A).

Both rhuSP-G in Tris buffer and in TA increased the wound closure rate in vitro. The addition of rhuSP-G resulted in increased gap closure by HCE cells in the presence of 100 μg/mL SP-G compared with 0 µg/mL. After 12, 24, and 48 h of incubation with 100 μg/mL SP-G, the remaining wound area was significantly smaller compared to controls (Table 1). A total of 10 μg/mL SP-G showed no effect on wound-healing rate. After 72 h, no effect could be detected even with 100 μg/mL SP-G.

In combination with TA, a significant pronounced wound-closure rate was observed after 12, 24, and 48 h incubation with 10 and 100 µg/mL SP-G compared to the TA control (Table 1). After 48 h incubation with SP-G + TA, the wound area was closed, whereas it took a longer time when incubating the cells with SP-G alone.

### 2.6. Effects of Cortisol on SP-G mRNA Levels

Cortisol is known for its stimulatory effect on surfactant protein synthesis [22]. We examined different concentrations of cortisol (0.1 µM, 1 µM, and 10 µM) and their effects on SP-G expression in cultured HCE cells. The concentrations of 0.1 µM and 1 µM of cortisol reflect similar levels to those found in stress conditions, and higher concentrations (10 µM) simulate pharmacological doses of glucocorticoids [23]. The use of cortisol leads to a significant increase in SP-G mRNA expression (Figure 4B). After incubation of the cells for 6 h, a significant increase (up to 15-fold) in mRNA expression was initially observed. After an incubation period of 24 h, the highest SP-G mRNA level was observed in the group with a concentration of 0.1 µM. It is noticeable that the effect of cortisol seems to decrease after 24 h, which was also observed after 48 h.

### 2.7. In Silico Molecular-Level Insight into Interactions of SP-G with Lipids of the Tear Film Lipid Layer (TFLL)

In molecular dynamics (MD) simulations, TFLL models with three different values of polar lipid packing were considered, with area per phospholipid (APPL) = ~0.7, 1.0, and 1.3 nm^2^. In each system, SP-G was initially placed in the aqueous subphase at ~2 nm distance from the water–lipid interface. In the two more packed systems (APPL = 0.7 and 1.0 nm^2^), the protein did not adsorb to the lipid film during 1 µs of the calculated MD trajectory. However, in the least packed system (APPL = 1.3 nm^2^), SP-G preferentially adsorbed at the water–polar 1-palmitoyl-2-oleyl-sn-glycero-3-phosphocholine (POPC) headgroup boundary within ~250 ns. After adsorption, during the following ~10 ns, the protein was partially incorporated in the lipid film and attained a stable orientation (Figure 5A) for the next 2.7 μs. The analysis of contacts between the protein and lipids (Figure 5E) revealed that polar, basic, and acidic residues of the adsorbed protein were, on average, interacting with water and polar POPC lipids. In contrast, the hydrophobic residues penetrated the POPC monolayer and interacted with nonpolar lipids, mostly with cholesteryl erucate (CE, Figure 5F,G). Spatial density profiles of molecular occupancy in the vicinity of the adsorbed protein (Figure 5H) demonstrate that adsorption of SP-G to the TFLL led to the formation of a pore in the POPC monolayer. The pore was occupied not only by the protein but also by water molecules that hydrated hydrophilic protein residues. Furthermore, some of the nonpolar lipids penetrated the formed pore and interacted with nonpolar residues.

In a separate MD trajectory, the system with adsorbed SP-G was laterally compressed from APPL = 1.4 to 0.3 nm^2^ in the course of 200 ns. During this simulation, the TFLL underwent significant structural alterations, with the formation of initial undulations of the polar POPC layer (Figure 5B), followed by the creation of invaginated structures (Figure 5C). Importantly, during this restructuring, SP-G remained stably adsorbed to the lipid film, and the integrity of the film (continuity of the polar and nonpolar layers) was not compromised (Figure 5B,C). In contrast, the TFLL without the protein, during the same lateral compression process, underwent a collapse forming inverse micelle-like structures, composed of POPC with water, which were pushed out from the polar layer to the nonpolar lipid phase (Figure 5D).

## 3. Discussion

The first description of a regulatory SP-G-mRNA, which locates in human chromosome 6p21.33, was achieved by Zhang et al. [24]. SP-G was demonstrated in developing lung tissue, bronchoalveolar lavage (BAL), and the immortalized alveolar type II cell line A549 [13,19]. Here, we demonstrate that SP-G is a component of the ocular surface and tears. It is produced by different tissues of the ocular system and the lacrimal apparatus, such as cornea, conjunctiva meibomian gland, and lacrimal gland. From these tissues, SP-G is secreted and becomes part of the tear film. We detected SP-G in corneal and conjunctival epithelial cells that are responsible for the formation of the mucous component of the tear film mainly in the form of membrane-bound mucins. However, we can only speculate that the SP-G produced here is also secreted into the tear film. Moreover, we detected SP-G in secretory duct cells as well as acinar cells of the lacrimal gland, which, together with the accessory lacrimal glands, form a major part of the aqueous component of the tear film (TF). Finally, we also localized SP-G in the meibomian gland acinar cells as well as cells lining the excretory duct, which are responsible for most of the superficial lipid component of the tear film. Since our Western blot and ELISA studies clearly show that SP-G is present in tear fluid, it is most likely that it is secreted into the tear film by all these tissues of the ocular surface (lacrimal gland, meibomian glands, conjunctiva, and cornea). In a previous study, we already demonstrated the presence of SP-G in various extrapulmonary tissues of the ocular surface but also other organs such as heart, kidney, sebaceous glands, and testis [13]. It is interesting to note that the SP-G concentration in human corneas is often up to four-fold higher than in the lacrimal gland tissue and conjunctiva. However, due to the lack of knowledge about the function of SP-G, this finding can only be speculated. However, a correspondingly higher concentration in the cornea was also detected for the protein PLUNC (Palate Lung Nasal Clone), which belongs to the family of surfactant proteins [25]. Here, it is speculated that PLUNC may have a function in the cornea, which is an organ with a very finely regulated fluid balance, in the context of fluid regulation or in immune defense. Similar functions are also conceivable for SP-G, especially since we were also able to detect it in the corneal endothelium, but remain speculative. Our Western blot analysis of tear fluid revealed distinct SP-G protein bands at 15 kDA. Considering that the protein might be posttranslational modified due to glycosylation, phosphorylation and palmitoylation, the distinct protein band at ≈15 kDa seems to represent the mature protein. This expression pattern (15 and 28 kDa) is in accordance with findings in human lung tissue, bronchoalveolar lavage (BAL), and the immortalized alveolar type II cell line A549 [13,19]. In tear fluid from patients with a hyperevaporative form of DED, our ELISA studies revealed a highly significant increase in SP-G concentration in the tear film. Since SP-G is also produced by meibocytes and the epithelial cells of the excretory ducts of the meibomian glands, it is probably integrated into the oily meibum during meibocyte differentiation. It has been clearly demonstrated that meibocytes produce various proteins that interact with the lipids of the meibum [8]. Interestingly, in this context, it appears that DED patients suffering from an evaporative form of DED also have increased SP-G concentrations. In these patients, less meibum is formed with an additional change in viscosity, so the increased SP-G concentration is rather surprising. Three hypotheses are conceivable for this: (1) More tears and SP-G are produced in response to increased evaporation in the absence of the protective lipid layer of the tear film in MGD, whereupon more SP-G is secreted by the lacrimal gland, conjunctiva, and cornea because of a possible immunologic function and to regulate the fluid balance of the ocular surface, resulting in a higher concentration of SP-G in the tear fluid. (2) A second conceivable reason for the increased SP-G concentration in the tear film is that the relative fluid volume in the tear film is greatly reduced and therefore the ratio of tear film components has shifted in favor of the SP-G protein or other proteins. (3) Another reason could be that SP-G may play a role in immune defense, as has been shown in particular for the surfactant proteins SP-A and SP-D, but also for SP-H [14,26,27]. Since in DED there is a subclinical to clinical inflammatory response at the ocular surface [28,29], SP-G could be upregulated here in response to inflammation. However, Mittal et al. reported that SP-G expression was downregulated by lipopolysaccharides in mice, which is contrary to the SP-G increase observed here in patients with evaporative form of the eye [19].

Our studies show that both HCE and HCjE cell lines we analyzed also express SP-G and thus can serve as in vitro models for SP-G studies. Immortalized cells in most cases deviate from the vital primary cells, also in the case of the used SV-40 HCE cells, which is a limitation of the study. This issue is known from numerous experiments and also other cells lines. But nevertheless, using these cells can provide first insights into to function of SP-G with respect to cellular function. For the experiments we used corneal epithelial cells for initial in vitro studies and performed stimulations with cortisol and wound healing experiments with recombinant SP-G. We chose cortisol as a stimulant because its physiological and pharmacological properties are well known, particularly in the context of maturation of the pulmonary surfactant system of premature infants [30,31]. Despite the commonly known assumption that cortisol might disturb wound healing, the effect of increased SP-G expression seems to exceed the influence of cortisol. After mechanical stress or wounding of confluent HCE cells, treatment with recombinant SP-G in combination with TA resulted in an increased rate of wound healing, such that specific lipids appear to promote the SP-G properties in question. The wound healing is presumably based on an increased migration of the cells through SP-G. An influence by proliferation cannot be excluded at present. Further investigations such as Ki-67 (proliferation marker) must follow. The molecular or even physical mechanisms behind this observation are still unclear and further experiments are needed to unravel this finding, but at least similar properties are known from SP-H [18]. The finding that recombinant SP-G reduces surface tension in tears, as shown by spinning drop analysis (Figure 3C), is further evidence for the function of SP-G. SP-G appears to be able to interact with lipids. The combination with TA enhances the effect and suggests that SP-G can only exert its full function in combination with specific lipids. TA is a phospholipid liposome spray with a high concentration of phosphatidylcholine 2-lysophosphatidylcholine, which improves the polar properties of the lipid layer (e.g., surface tension and solubility) and enhances the distribution of the lipid in the tear film [22,32,33]. First lipid simulations already pointed out the interaction with lipids [21,34]. Already Krieger et al. mentioned in an analysis of a water box that palmitoylation of SP-G can act as a membrane anchor in the lipid environment to stabilize the protein on the lipid surface or mediate its adsorption to the membrane by penetration into the hydrophobic protein core. Therefore, we hypothesize that SP-G in the tear film may reduce surface tension and stabilize the tear film.

In order to obtain deeper insights into whether and how SP-G may potentially interact with tear film lipids, we performed MD simulations. MD simulations revealed molecular-level details of SP-G interactions with an in silico model of TFLL. SP-G adsorbs, interacts with lipids, and incorporates in TFLL with low packing of polar lipids. In the tear film, the packing of lipids at the tear–air interface varies with time as the tear film undergoes significant restructuring, involving thinning, breaking, and respreading after blinks [35]. Furthermore, the blinks cause mechanical mixing of the lipid material with the aqueous subphase. Hence, it can be assumed that SP-G adsorbs to TFLL in the actual tear film. Simulations also show that SP-G remains stably incorporated upon adsorption despite the lateral mechanical stress. The adsorption of SP-G observed at the molecular level is in accord with previous MD simulations, which considered SP-G interacting with purely polar lipid films [21].The hydrophilic and hydrophobic residues of the adsorbed SP-G interact predominantly with polar and nonpolar lipid molecules, respectively. After incorporating to the lipid film, SP-G spans from the aqueous subphase, through the polar monolayer layer, to the nonpolar lipid layer. To a certain extent, the hydrophilic residues stay hydrated, with the hydrating water partially penetrating the lipid environment in the vicinity of the protein. The adsorption and interaction with lipids reported at the molecular scale by MD are in accord with the experimentally observed surface tension reduction by SP-G. Interactions with POPC lipids are dominant, which agrees well with the observed enhanced activity of SP-G in the presence of phospholipids in tear substitutes. Importantly, MD revealed that the incorporation of SP-G to TFLL enhances the structural stability of the lipid film, allowing it to withstand higher lateral pressure than the TFLL without the protein. In the absence of SP-G, a collapse was observed under corresponding lateral pressure. The present simulations point to an essential role of both polar and nonpolar lipids for SP-G embedment. Of note, the previous computational models of SP-G-lipid interactions considered only polar lipids. It should also be mentioned that no posttranslational modifications of SP-G were considered here; as shown in previous studies, such modifications can additionally enhance SP-G interactions with lipids [21].

In summary, we identified SP-G as a secretory protein at the ocular surface and in tear fluid. There it has been shown to be involved in wound healing of the corneal epithelium. In addition, SP-G is able to reduce the tension of the tear film, especially of tear fluid from patients suffering from dry eyes. Molecular simulations show that SP-G specifically interacts with polar and nonpolar components of the tear lipid film and increases its mechanical stability. Further studies are needed to gain deeper insights into the function and mode of action of SP-G at the ocular surface, particularly with regard to SP-G on the ocular surface immune system and in relation to the pathogenesis and potential treatments of DED and meibomian gland dysfunction.

## 4. Materials and Methods

### 4.1. Generation of Samples

Tissue samples were obtained from 6 body donors (3 men, 3 women; age range: 63–95 years) donated to the Institute of Functional and Clinical Anatomy, FAU Erlangen-Nuremberg, Erlangen, Germany. The study was approved according to institutional review board regulations and is consistent with the goals of the Declaration of Helsinki. Samples were collected from the body donors within 24 h post mortem. Half of each sample was immediately frozen at −80 °C for biological studies. The other half was fixed in 4% paraformaldehyde for subsequent embedding in paraffin. Bronchial mucosa and lung tissue were used as positive controls.

The procedure of collecting tear fluid as well as the clinical diagnosis of dry eye has already been described and performed in detail by Schicht et al. [25]. Tear fluid samples were collected with Schirmer strips through PD (Dr. Jacobi at the Department of Ophthalmology, Friedrich Alexander University Erlangen-Nuremberg, Germany). These specimens were obtained in compliance with good clinical practice and with informed consent. Ethical approval was obtained from the Ethics Committee of the University of Erlangen-Nuremberg (54-2532.1-35/13). Written consent was received from all patients and subjects after explanation of the procedures and study requirements. Tears from 20 control samples were included in the study. They showed no symptoms for dry eye disease or ocular discomfort, did not use any artificial tears or lubricant eye drops, and did not suffer from any autoimmune disorders or other eye diseases, including ocular allergies, and had no history of eye surgery or contact lens wearing. A total of 40 tear fluid samples of 40 patients (mean age 42.8 ± 8.3 years) with dry eye (DEWS dry eye severity level 2) were also enrolled in the study. Inclusion criteria were as follows: (i) Ocular Surface Disease Index questionnaire Score (OSDI Score) > 40, (ii) tear break-up time (TBUT) ≤ 10 s, (iii) Schirmer test with anesthesia ≤ 10 mm, (iv) lid-parallel conjunctival folds (LIPCOF) > 2. A more detailed description of these points is given below. For the ELISA, the study of patients was limited to the most common form, i.e., evaporative dry eye due to meibomian gland dysfunction (EDE, *n* = 40). EDE patients had a tear break-up time of ≤5 s and a Schirmer test ≤ 10 mm. Exclusion criteria consisted of a medical history of trauma or infection, ocular allergies, pregnancy, lactation, history of refractive surgery/ocular surgery/any other surgery within the previous 6 months, immunosuppressive medications, or the use of contact lenses within 14 days prior to ophthalmological examination. Moreover, patients wearing punctum plugs, patients with history or evidence of epithelial keratitis derived from herpes simplex infection, recent varicella infection, corneal or conjunctival viral disease, acute corneal, conjunctival, or palpebral bacterial infection, or ocular fungal infection, were excluded from this study.

### 4.2. Cell Culture

Immortalized HCE and HCjE cells were cultured as previously described by [18]. SV40-transformed human corneal epithelial cells (HCE cells, obtained from Kaoru Araki-Sasaki, Tane Memorial Eye Hospital, Osaka, Japan, passage number 18–27) [36] as well as a human spontaneously immortalized epithelial cell line from normal human conjunctiva [IOBA-NHC, here referred to as HCjE cells, obtained from Yolanda Diebold, University Institute of Applied Ophthalmobiology (IOBA), University of Valladolid, Valladolid, Spain] [37] were cultured as monolayer and used for further stimulation experiments. For stimulation experiments, cells (1 × 10^6^) were seeded in Petri dishes and cultured until confluence was reached. Before cells were treated with cortisol (0.1 µM, 1 µM, and 10 µM) (Sigma-Aldrich, St. Louis, MO, USA), they were washed with phosphate-buffered saline (PBS) and incubated with serum-free medium for at least 6 h. They were respectively collected after incubating for 6 h, 12 h, 24 h, and 48 h, then isolated for RNA extraction and for continuing further qPCR analysis.

### 4.3. mRNA Extraction and cDNA Synthesis

Total mRNA was extracted from cell cultures as well as from human tissues: lacrimal gland (*n* = 12), eyelids (*n* = 12), conjunctiva (*n* = 12), and cornea (*n* = 12) using peqGOLD TriFast reagent (VWR International GmbH, Darmstadt, Germany). For mechanical comminution, SpeedMill plus (Analytik Jena AG, Jena, Germany) was used. After centrifugation (10 min, 15,520× *g*), the supernatant was applied for mRNA isolation. DNA contaminations were removed using a standard protocol. Reverse transcription of mRNA samples into first-strand cDNA was performed using the RevertAid Reverse Transcriptase Kit (Thermo Fischer, Waltham, MA, USA) according to the manufacturer’s protocol. Two micrograms of total RNA were used for each reaction to cDNA. Gene-specific intron-spanning primers previously synthesized at MWG Biotech AG (Ebersberg, Germany) were used for PCR.

### 4.4. Polymerase Chain Reaction (PCR)

RT-PCR was performed according to the following standard protocol. Primers used for conventional PCR: SP-G sense (5′- AGCGTGAGCAGGAAGGTTCT -3′) and antisense (5′- GCGCCATGTAAGAGAGCTCT -3′) were used for qRT PCR (i.e., real time). To estimate the amount of amplified PCR product, a β-actin PCR was performed with specific primers: sense (5′-GAT CCT CAC CGA GCG CGG CTA CA-3′) and antisense (5′-GCG GAT GTC CAC GTC ACA CTT CA-3′). The reference gene β-actin was used as an internal control for assessing the integrity and stability of the transcribed cDNA.

### 4.5. Quantitative Real-Time RT-PCR

Gene expression was analyzed with quantitative Real-Time RT-PCR (qPCR) using a LightCyler480^®^ system (Roche) as previously described by Schicht et al., 2018 [18]. The PCR reaction mixture contained 10 μL LightCycler480^®^5x probe mastermix, 0.25 μL of each primer and 2 μL of each cDNA, 0,4 µL Universal ProbeLibrary probe #13 (UPL, Roche) for SFTA2 or #22 for 18S (10 µM) and 7,1 µL nuclease free water. On each 96-well plate, qPCR was performed with a cycle of 5 min at 95 °C, 55 cycles of 15 s at 95 °C, 30 s at 60 °C and 1 s at 72 °C, to confirm amplification of specific transcripts. SP-G and 18 s (sense 5′-GGT GCA TGG CCG TTC TTA-3′ and antisense 5′-TGC CAG AGT CTC GTT CGT TA-3′) primers as well as the corresponding UPL probes (see above) were generated by using the ProbeFinderTM software (Version 2.04, Roche). A standard curve was generated by serial dilutions of cDNA from non-stimulated cells. To standardize mRNA concentration the transcript levels of the housekeeping gene small ribosomal subunit (18S rRNA) were determined in parallel for each sample, and relative transcript levels were corrected by normalization based on the 18S rRNA transcript levels. Each sample was performed in triplicate, and the changes in gene expression were calculated by applying the ΔΔCt method.

### 4.6. Western Blot

The Western blot aimed to detect SP-G in tear fluid from patients suffering from DED and healthy donors. The tear fluid was mixed with RSB buffer (consisting of Glycerin, 10% SDS, Tris·Base and Mercaptoethanol) and heated at 95 °C for 5 min before loading. Considering the molecular weight of samples, 15% SDS-polyacrylamide gel for electrophoresis was chosen to separate the protein. Afterwards, separated proteins were transferred into a nitrocellulose membrane using a Semi-Dry Transfer Cell (BIO-RAD). Membrane was incubated with the rabbit anti-SP-G antibody in 1% BSA-PBST (1:200) (ABBEXA, Cambridge, UK) at 4 °C overnight and anti-rabbit-HRP in 5% BSA-PBST (1:1000) (Cell Signaling, Danvers, MA, USA) at room temperature for 2 h after blocking with 5% BSA in 1xPBST. Then, the bands were detected with ECL Western blotting detection reagents (EMD Millipore Corporation, Burlington; MA, USA). The molecular weights of the detected protein bands were estimated by using the standard protein marker, Precision Plus Protein Dual Color Standards (ranging from 10 to 250 kDa).

### 4.7. Immunohistochemistry

Immunohistochemical analysis was performed as previously described by [38]. Tissue samples of lacrimal gland, eyelids, conjunctiva, and cornea from donors were embedded in kerosene, sectioned (5 μm), and deparaffinized for immunohistochemical staining.

Immunohistochemistry was performed according to a standard general protocol. Sections were incubated with primary polyclonal antibodies (1:50) (Abbexa) against SP-G overnight at 4 °C, and secondary antibodies (goat anti-rabbit) (1:200) (Dako) were incubated for at least 2 h at room temperature. Visualization was performed with AEC (Dako) for an appropriate time.

For immunohistochemical analysis of cells, cells were cultured on glass coverslips and fixed with 4% PFA for 1 min. Visualization of antibody binding was performed with AEC for an appropriate time. Sections were counterstained with hematoxylin and then coverslipped with Aquatex (Merck kGaA, Darmstadt, Germany).

For fluorescent staining, cells were incubated overnight with rabbit anti-SP-G antibody (1:50 in TBST) (Seqlab) and for 2 h with fluorescein isothiocyanate (FITC)-conjugated antibody (Alexa 488, green) diluted 1:200 with TBS. Cells were embedded with DAPI-glycerol (PBS-glycerol 1:1, by adding 10 μL of a 2 mg/mL stock DAPI solution) on glass slides. The slides were examined using a Keyence Biorevo BZ9000 microscope.

### 4.8. Enzyme-Linked Immunosorbent Assay (ELISA)

Tear fluid of healthy donors and from patients suffering from DED (hyperevaporative) as well as ocular tissue samples were analyzed by quantitative sandwich ELISA. The analysis was performed by using SFTA2 ELISA kit and the relevant protocols from Cusabio Biotech Co., Ltd. (Wuhan, China). Quantification was accomplished by comparison with the 2-fold standard dilution series and the determined values for antigen concentration ranging from 0.312 ng/mL to 20 ng/mL. Subsequently, each sample was approximated to ng/mg.

### 4.9. In Vitro Wound Healing Scratch Assay

HCE cells were seeded in Petri dishes and cultured until confluence was reached. Before scratching, the cells were incubated with serum-free medium (sfm) for at least 6 h. Then, the cell lawn was disrupted with a pipette tip (100 µL yellow tip, Eppendorf) to create an epithelial defect. Subsequently, cells were stimulated with 10 or 100 ng/mL rhuSP-G in combination with a tear replacement solution (phospholipid liposome spray—Tears Again (TA) Optima Pharmazeutische GmbH, Hallbergmoos, Germany) in sfm. TA is a phospholipid liposome spray with a high concentration of phosphatidylcholine 2-lysophosphatidylcholine, which improves the polar properties of the lipid layer (e.g., surface tension and solubility) and improves the distribution of the lipid in the tear film [32,33,34]. TA was purchased at a local pharmacy. Optima GmbH did not financially support the investigations in this study. Tris (Tris(hydroxymethyl)aminomethane) buffer were used to stabilize the SP-G protein and as an internal control (sfm + Tris) to exclude the influence of Tris buffer. Subsequently, the wound edges were measured at different time points (0 h, 3 h, 6 h, 12 h, 18 h, 24 h, 36 h, and 48 h). ImageJ (open source software) was used to evaluate the wound area. The average distance was calculated and analyzed to evaluate the effect of SP-G.

### 4.10. Interfacial Tension

The experiment was performed as previously described by [18]. The method itself is described by www.kruss.de (accessed on 17 May 2022). A spinning drop interfacial tensiometer (model Kruss, Germany) was used to measure the surface and interfacial tensions of the tear fluid containing various concentrations of SP-G at room temperature in combination with Tears Again (TA) (see above).

### 4.11. Molecular Dynamics Simulations

Molecular dynamics simulations were performed employing the setup similar to that used in previous tear film computational studies [39,40]. In short, in the simulation box elongated along one direction, a slab of water was placed forming two water–air interfaces. At each interface, a lipid film consisting of POPC (1-palmitoyl-2-oleyl-sn-glycero-3-phosphocholine), BO (behenyl oleate), and CE (cholesteryl erucate) was spread, and the system was equilibrated for a few hundred nanoseconds. One molecule of SP-G was then placed in the water slab and its adsorption was simulated in a direct MD simulation. The MARTINI coarse grain model was used for describing molecular interactions, with the elastic network method used to maintain the shape of the protein [41]. The slab consisted of ~90,000 water beads and the lipid film at each interface comprised of 727 molecules of POPC, and 1250 molecules of each BO and CE. The size of the simulation box was equal to 22 × 22 × 104 nm^2^ and the corresponding area per polar POPC was equal to 0.7 nm^2^. Less laterally packed interfaces were simulated be reducing the number of POPC lipids, equally at both interfaces. The size of the simulation box was kept constant, and the temperature was set to an average 310 K (NVT ensemble). Simulations of lateral film compression were realized by turning on the pressure coupling algorithm with the lateral pressure of 15 bar. Several replicas of each simulation were performed. GROMACS (ver. 2020.4) software was employed as the MD simulation engine and VMD (ver. 1.9.3) was used for visualization [42,43].

### 4.12. Statistical Analysis

Calculations and visualizations were performed using GraphPad Prism 6 (GraphPad Prism Software). The statistical analysis was carried out by using an unpaired, a two-sided Mann–Whitney U-test (nonparametric data) or two-sided Welch’s t-test (parametric data) with mean ± standard error of the mean (SEM) of all data. Significance value was defined at * *p* < 0.05, ** *p* < 0.005 or *** *p* < 0.0005.

## Figures and Tables

**Figure 1 ijms-23-05783-f001:**
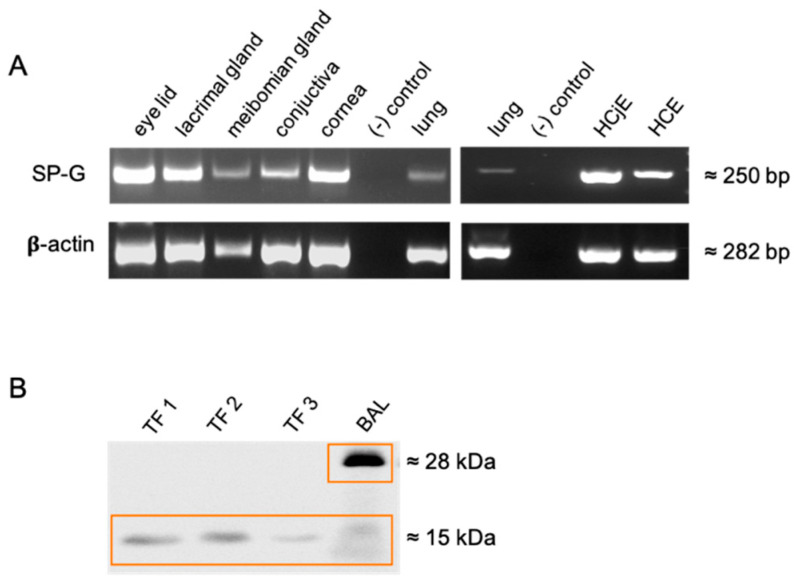
(**A**) Detection of SP-G mRNA in the following samples: lacrimal gland, meibomian gland, eyelid, conjunctiva, cornea, human corneal epithelial cells (HCE), human conjunctival epithelial cells (HCjE). Samples without cDNA were used as negative controls, whereas samples containing lung cDNA were used as positive controls. (**B**) Western blot analysis with a polyclonal anti-rabbit SP-G antibody incubated on tear film (TF) (TF 1–3) and bronchoalveolar lavage (BAL) as positive control.

**Figure 2 ijms-23-05783-f002:**
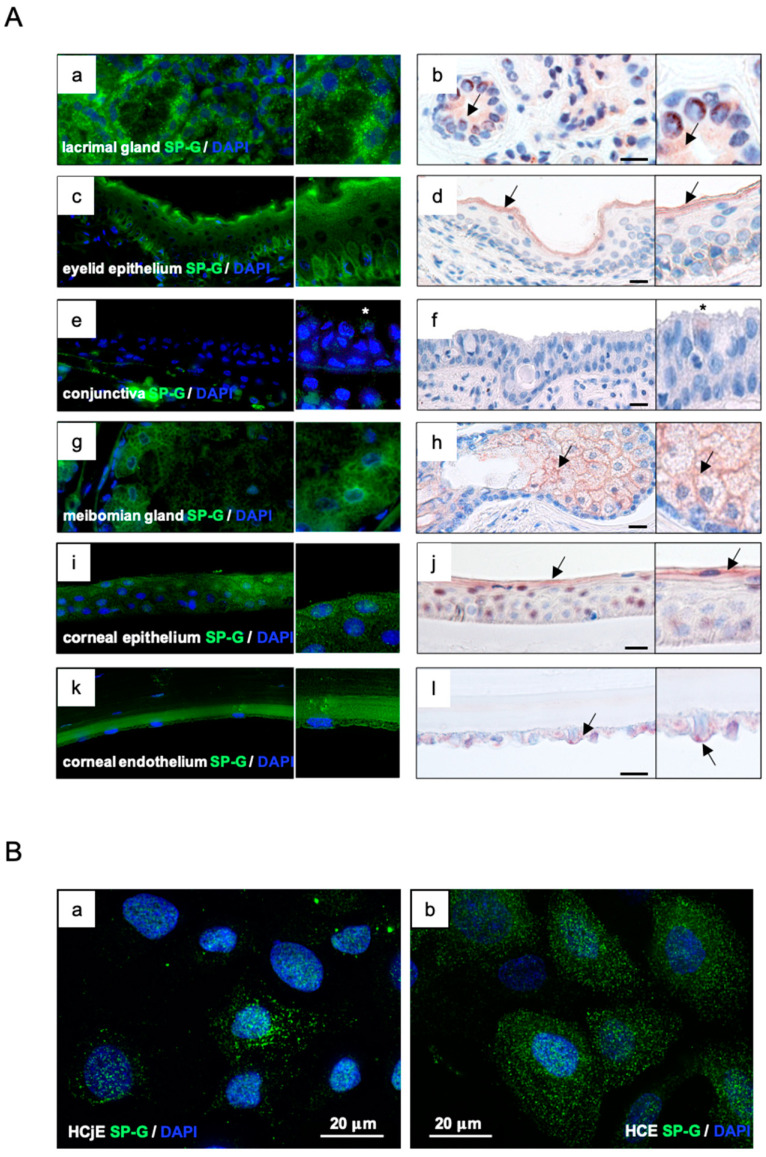
(**A**) Immunohistochemical detection of SP-G (green and red) in lacrimal glands, meibomian glands, eyelid, conjunctiva, corneal epithelium, and corneal endothelium by fluorescence (green, left column) and AEC substrate staining (red, black arrow, right column). (**a**,**b**) Lacrimal glands: detection in cells of secretory ducts as well as in serous acini of the gland. (**c**,**d**) Eyelid: SP-G was detected in the basal epithelium of the eyelid and protein distribution as a superficial layer. (**e**,**f**) Conjunctiva: multilayered epithelium of the conjunctiva from healthy donors shows weak reactivity (white/black star). (**g**,**h**) Meibomian glands: SP-G detected within the cytoplasm of the gland and the excretory duct system cells. (**i**,**j**) Cornea epithelial: protein detected in the basal epithelium and superficial layers of epithelial cells. (**k**,**l**) Cornea endothelial: SP-G detected in the basal epithelial cells. (**B**) HCE (**a**) and HCjE (**b**) cell lines: cytoplasmatic localization of SP-G red and green (SP-G = green; DAPI (4′,6-diamidino-2-phenylindole) = blue). The scale bar equal 50 µm (**A**) and 20µm (**B**).

**Figure 3 ijms-23-05783-f003:**
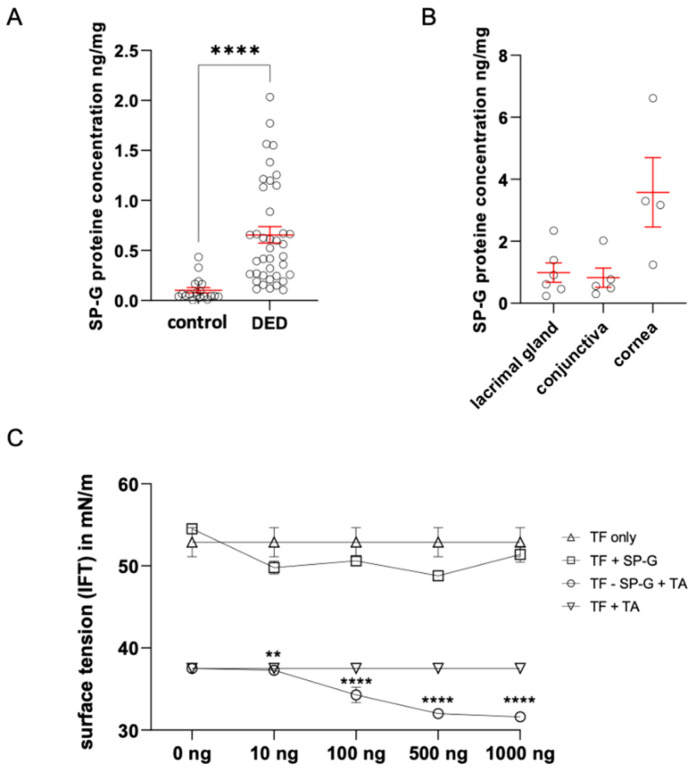
Protein quantification of Human Putative Protein SP-G (SFTA2) in tear fluid and interfacial tension (IFT) analysis. (**A**) Quantification of SP-G in TF. Mean values (red) of healthy subjects: 0.152 ng/mg, EDE: 0.641 ng/mg. (**B**) Quantification of SP-G in ocular tissue samples. Mean values of lacrimal gland: 0.991 ng/mg, conjunctiva: 0.862 ng/mg and in cornea 3.578 ng/mg. (**C**) IFT for different rhu-SP-G concentrations in tear fluid samples. As control we used TF only and TA with TF. The IFT is expressed in mN/m and as mean ± SEM. Statistical significance: ** *p* ≤ 0.005, **** *p* ≤ 0.0005.

**Figure 4 ijms-23-05783-f004:**
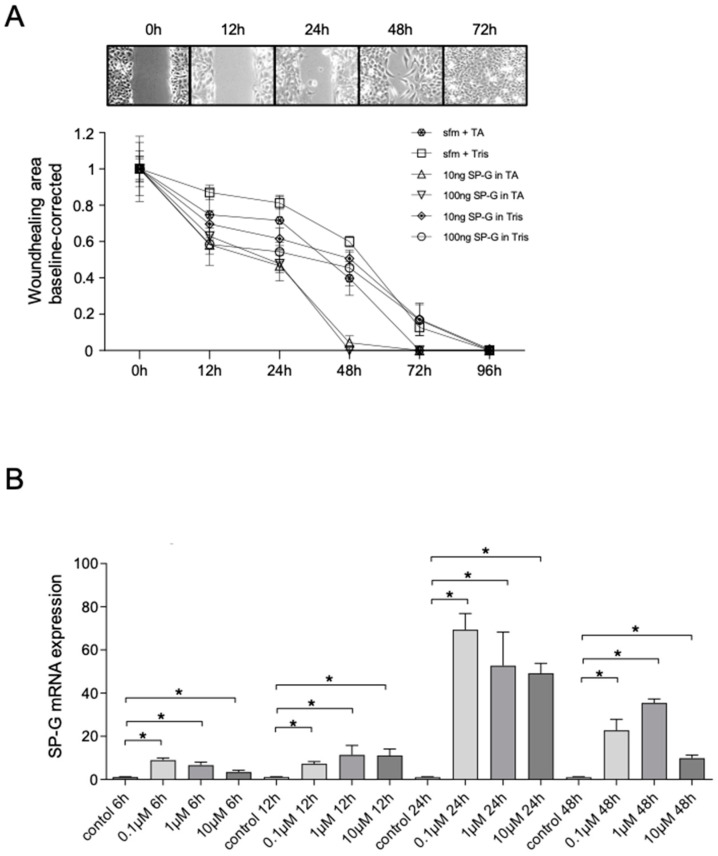
In vitro wound healing scratch assay with HCE cells. (**A**) Baseline corrected wounded area in HCE monolayer cultures incubated with 10 and 100 µg/mL SP-G in TA were compared to the control group only treated with TA and with 10 and 100 µg/mL SP-G in tris buffer were compared to the control group with pure tris buffer. The wound-healing rates (closure of the scratch) were significantly higher during stimulation with rhuSP-G, combined to TA the effect was more obvious. After 48 h of incubation with 100 µg/mL rhuSP-G and combined with TA, the wound area is already closed. The wound-healing distance was expressed as mean ± SEM. (**B**) Stimulation of HCE cells for 6, 12, 24, and 48 h after treatment with 0.1, 1, and 10 µM cortisol, respectively. SP-G mRNA expression of HCE cells after stimulation with cortisol was determined by qRT-PCR. SP-G mRNA expression levels are expressed as mean ± SEM. (*n* = 4; * *p* ≤ 0.05).

**Figure 5 ijms-23-05783-f005:**
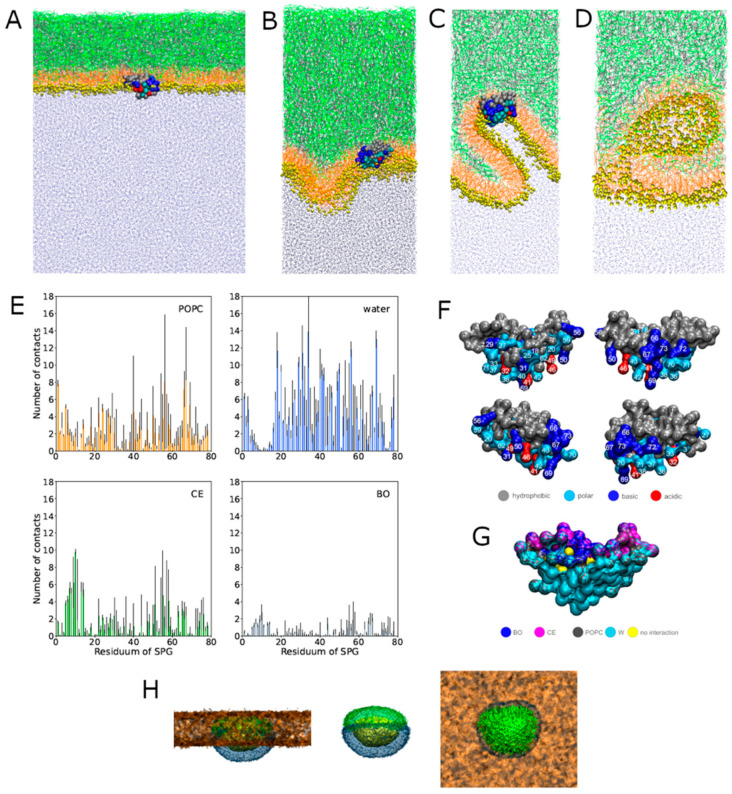
In silico MD simulations of SP-G interactions with a TFLL model. (**A**–**C**) Representative snapshots of the simulation boxes showing SP-G interacting with TFLL. (**D**) A representative snapshot of the collapsed TFLL in the absence of SP-G. Color coding in all snapshots: POPC lipids—orange lines; polar head groups of POPC—yellow spheres; CE—green lines; behenyl oleate (BO)—gray lines; water—blue dots; SP-G represented by spheres with coloration dependent on the residue type (hydrophobic—gray; polar—blue; basic—dark blue; acidic—red). (**E**) Contact numbers (within the 0.7-nm cutoff) between SP-G and POPC, water, CE, and BO; error bars calculated by block averaging. (**F**) Visualization of the character (hydrophobic, polar, basic, and acidic) of the SP-G residues; four different orientations of the SP-G protein are shown for clarity. (**G**) Visualization of the contact residues (within the 0.7-nm cutoff) of SP-G with individual lipid types and water in the equilibrated system (as shown in panel **A**). (**H**) Spatial densities of SP-G (yellow), POPC (orange), water (blue), and CE (green) in the equilibrated system (as shown in panel **A**). The density of all POPC groups is shown, while only the density of the closely interacting molecules is shown for water and CE. A side view of the film is shown on the left, the same view without POPC is shown in the middle, and the top view (from the air side) is shown on the right.

**Table 1 ijms-23-05783-t001:** Results of the statistical analysis of the scratch assays with the individual pairings. ** *p* ≤ 0.005, *** *p* ≤ 0.0005.

Time Points	Pairings	Statistical Significance
12 h	sfm + TA vs. 100 ng SP-G + TA	***
12 h	sfm + Tris vs. 100 ng SP-G + Tris	***
24 h	sfm + TA vs. 10 ng SP-G + TA	***
24 h	sfm + TA vs. 100 ng SP-G + TA	***
24 h	sfm + Tris vs. 100 ng SP-G + Tris	**
48 h	sfm + TA vs. 10 ng SP-G + TA	**
48 h	sfm + TA vs. 100 ng SP-G + TA	***

## Data Availability

Please contact authors for data requests (M.S.; email: martin.schicht@fau.de).

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
