# Peer review of "The Potential Role of SP-G as Surface Tension Regulator in Tear Film: From Molecular Simulations to Experimental Observations"

_ijms, 2022, doi:10.3390/ijms23105783_

Round 1

Reviewer 1 Report

The authors attempted to describe the detection, characterization, and putative role of surfactant protein G (SP-G/SFTA2) with respect to wound healing and surface activity. This was an interesting study, but there were several questions to be answered and discussed.

In the introduction or discussion session, the authors needed to review previous studies regarding other surfactant proteins.

In the method, the authors needed to declare how they gathered tears from the subjects in detail. Moreover, I wondered if their study was registered in a public trials registry, because they seemed to prospectively register dry eye patients and normal controls.  

In the figure 1, the authors normalized the levels of SP-G with beta-actin. They needed to show the ratio of SP-G/beta-actin as a separate figure, because the beta-actin levels were quite different among tissues.

In the abstract, the authors described that “In dry eye disease (DED) patients, increased expression of SP-G and accelerated wound healing were demonstrated.” Considering their outcomes, it was not showed that the DED patients had accelerated wound healing, which needed to be revised.

In respect to wound healing, the authors’ findings did not show how the SP-G could help the epithelial wound healing. They checked only migration of corneal epithelial cells in vitro using an epithelial scratch assay. They needed to additionally perform proliferation and differentiation assays to know how. Moreover, the authors needed to perform in vivo mice study using corneal epithelial debridement model to confirm their findings.

The cortisol was well-known to disturb epithelial wound healing in the clinical setting. Conversely, the authors describe the use of cortisol led to a significant increase in SP-G mRNA expression, which seemed to help wound healing. Because there could be confusion, they needed to more clearly show the beneficial effect of cortisol in epithelial wound healing.  

Author Response

Point-by-point response to the reviewers' comments

Reviewer #1:

  1. In the method, the authors needed to declare how they gathered tears from the subjects in detail. Moreover, I wondered if their study was registered in a public trials registry, because they seemed to prospectively register dry eye patients and normal controls.

Answer: We thank the reviewer for this comment. In our methods section, we have included the last manuscript describing the sample preparation in detail. To avoid violating the rights of the journal, we have only cited the article where the reader can learn details about the sample collection.

To ensure that this information is not lost in the reading, we have changed the sentence to make it more prominent, line 366

  1. In the figure 1, the authors normalized the levels of SP-G with beta-actin. They needed to show the ratio of SP-G/beta-actin as a separate figure, because the beta-actin levels were quite different among tissues.

Answer: We thank the reviewer for this advice. We have repeated and improved the PCR. Figure 1

  1. In the abstract, the authors described that “In dry eye disease (DED) patients, increased expression of SP-G and accelerated wound healing were demonstrated.” Considering their outcomes, it was not showed that the DED patients had accelerated wound healing, which needed to be revised.

Answer: We thank the reviewer for this tip. We have modified the abstract, line 28 - 33

  1. In respect to wound healing, the authors’ findings did not show how the SP-G could help the epithelial wound healing. They checked only migration of corneal epithelial cells in vitro using an epithelial scratch assay. They needed to additionally perform proliferation and differentiation assays to know how. Moreover, the authors needed to perform in vivo mice study using corneal epithelial debridement model to confirm their findings.

Answer: We thank the reviewer for this comment. It is clear that the scratch assay is not sufficient to guarantee an informed conclusion, but it should provide a first insight to drive further research. Our research group has many years of experience studying corneal wound healing (Wittmann et al. 2018 scientific reports) and other tissues.

It should not be underestimated how difficult it is to obtain approval for animal experiments in Germany, which is extremely complex from both an ethical and research perspective. This is a first description of a new protein and there will be further research approaches. We have pointed out in the discussion section that this is a migration study, line 315.

  1. The cortisol was well-known to disturb epithelial wound healing in the clinical setting. Conversely, the authors describe the use of cortisol led to a significant increase in SP-G mRNA expression, which seemed to help wound healing. Because there could be confusion, they needed to more clearly show the beneficial effect of cortisol in epithelial wound healing.

Answer: We thank the reviewer for this comment. Despite the common assumption that cortisol might disturb wound healing, the effect of increased expression of SP-G is exceeding the influence of present cortisol. We stated this issue in the revised version of the manuscript, line 310.

Reviewer 2 Report

Line 61 :  "a role extrapulmonary"  should be changed to "an extrapulmonary role"

Line 62:  "especially also at the ocular surface" can be changed to "including the ocular surface"

Figure 1A:  the positive control bands (lung cDNA) are the faintest bands in the PCR.  This makes me worry that it is possible that the SP-G in the lung and SP-G in the eye may be different.  Also, the B-actin loading control bands are of various quantities.  This is an issue that needs to be fixed.

Figure 1B:  There is no loading control in this Western Blot.  Also need to address why there is expected 28kDa band for SP-G in lung but not eye.

Figure 2:  There are many issues with this figure and legend.  What is the difference between column 1 and 2?  Is the right column an H&E stain?  What is meant by red staining?  The fluorescence is green.  It makes it confusing for reader.  The legend mentions black arrows, but I see no black arrows in figure.  There is an asterisks in panels c,f but no mention of the purpose of the asterisk in the legend nor in the results section 2.2.

Section 2.3.  If these quantifications were done by ELISA, that needs to be mentioned in first sentence.  Line 142 states "Fig 2a" but I believe this should be FIG 3B.

Figure legend 3.  there is incosistency between using decimals vs. commas in reporting values.  example: line 149 "0,872 ng/mg " vs. line 141 "0.826 ng/mg"

Line 159:  "IFT from 46 to 25 mN/m-1"  the graph FIG 3c has no data points within these value ranges.

Line 160 Fig 3b should be Fig 3c.

Section 2.6 needs MAJOR WORK.  the results described in this section do NOT match up with what is in the figure.

Figure 4 Text is too small. 

Figure 4 legend shows no difference in symbols between sfm + TA and 100ng SPG in TRIS.  both use open circles.

Figure Legend 4A states these are percentages of wounded area (not percentages) then later states wound healing distance expressed as mean +/- SEM without units.

Figure 4B.  can the cortisol be titrated down to where there is no longer an effect?  it seems to be signifacnt at .1, 1, and 10 micromolar.  What happens at .01? .001?

Section 2.7.  There are strange spaces in front of some of the values presented.  Also repeated use of nm2 as a unit.  is this meant to be nm2

Author Response

Point-by-point response to the reviewers' comments

Reviewer #2:

  1. Line 61: "a role extrapulmonary" should be changed to "an extrapulmonary role"

Answer: The sentence has been changed accordingly, line 61.

  1. Line 62: "especially also at the ocular surface" can be changed to "including the ocular surface"

Answer: The sentence has been changed accordingly, line 62.

  1. Figure 1A: The positive control bands (lung cDNA) are the faintest bands in the PCR. This makes me worry that it is possible that the SP-G in the lung and SP-G in the eye may be different. Also, the B-actin loading control bands are of various quantities. This is an issue that needs to be fixed.

Answer: We thank the reviewer for this advice. We have repeated and improved the PCR. The fact that SP-G seems to be less expressed in the lung is due to the fact that SP-G is mainly present in the brochal tissue of the lung and it is not always clear in the surgical material how much brochal tissue is present in the total tissue.

  1. Figure 1B: There is no loading control in this Western Blot. Also need to address why there is expected 28kDa band for SP-G in lung but not eye.

Answer: We thank the reviewer for this comment. We did not use a classical loading control, because most of the loading controls in the tear film can be different in the subjects. Reasons for this are allergies, time, but also pathological reasons that are not diagnosed until then. The Western blot in Figure 1 was used only to detect the protein and quantification was then done by ELISA.

We then added the comment of why we only have the 28kDa in the lung in the discussion, line 276.

  1. Figure 2: There are many issues with this figure and legend. What is the difference between column 1 and 2? Is the right column an H&E stain? What is meant by red staining? The fluorescence is green. It makes it confusing for reader. The legend mentions black arrows, but I see no black arrows in figure. There is an asterisks in panels c,f but no mention of the purpose of the asterisk in the legend nor in the results section 2.2.

Answer: We thank the reviewer for this comment. We have revised the figure and the corresponding text sections, line 103-105 and legend.

  1. Section 2.3. If these quantifications were done by ELISA, that needs to be mentioned in first sentence.

Answer: We thank the reviewer for this comment, we have added the note on ELISA, line 135.

  1. Line 142 states "Fig 2a" but I believe this should be FIG 3B.

Answer: We thank the reviewer for this comment.

  1. Figure legend 3. there is incosistency between using decimals vs. commas in reporting values. example: line 149 "0,872 ng/mg " vs. line 141 "0.826 ng/mg"

Answer: We thank the reviewer for this comment, we have adjusted the values.

  1. Line 159: "IFT from 46 to 25 mN/m-1" the graph FIG 3c has no data points within these value ranges.

Answer: We thank the reviewer for this comment, we have revised the graph.

  1. Line 160 Fig 3b should be Fig 3c.

Answer: We thank the reviewer for this comment.

  1. Section 2.6 needs MAJOR WORK. the results described in this section do NOT match up with what is in the figure.

Answer: We thank the reviewer for this comment, we have revised this section.

  1. Figure 4 Text is too small.

Answer: We thank the reviewer for this comment, we have create a table for this section.

  1. Figure 4 legend shows no difference in symbols between sfm + TA and 100ng SPG in TRIS. both use open circles.

Answer: We thank the reviewer for this comment, we apologize for this mistake.

  1. Figure Legend 4A states these are percentages of wounded area (not percentages) then later states wound healing distance expressed as mean +/- SEM without units.

Answer: We thank the reviewer for this comment, we have revised this section.

  1. Figure 4B. can the cortisol be titrated down to where there is no longer an effect? it seems to be significant at .1, 1, and 10 micromolar. What happens at .01? .001?

Answer: We thank the reviewer for this comment. We used cortisol concentrations that are known to have pharmacological effects or at least concentrations that are present in stress situations. We stated this point now within the manuscript.

  1. Section 2.7. There are strange spaces in front of some of the values presented. Also repeated use of nm2 as a unit.  is this meant to be nm2?

Answer: We thank the reviewer for this comment, we apologize for this mistake. Several special characters were not displayed correctly and also the nm in the square was not clear.

Round 2

Reviewer 2 Report

Lines 157-160 read.  "A concentration of 100 ng/ml SP-G resulted in a significant decrease in tear film surface tension (IFT) from 46 to 25 mN/m (n = 3). The other two samples used as loading controls showed no effect (Fig. 3C)"  However, the graph in figure 3C does not support the claim that IFT decreases from 46 to 25 mN/m as the y-axis only goes down to 30 mN/m.

There is a label missing in figure 1 PCR between lacrimal gland and conjuctiva.  This lane also has a much less amount of b-actin in the control band.  In the original manuscript this was labeled as "meibomian gland" but was never mentioned in the text nor figure legend of this figure.  If it isn't needed, omit it.

The authors addressed the rest of the issues of the manuscript appropriately. 

Author Response

Reviewer #2:

We thank the reviewer 2 for his advice and assistance in revising the paper.

  1. Lines 157-160 read.  "A concentration of 100 ng/ml SP-G resulted in a significant decrease in tear film surface tension (IFT) from46 to 25mN/m (n = 3). The other two samples used as loading controls showed no effect (Fig. 3C)"  However, the graph in figure 3C does not support the claim that IFT decreases from 46 to 25 mN/m as the y-axis only goes down to 30 mN/m.

Answer: We thank the reviewer for this advice. We have revised the corresponding text sections.

  1. There is a label missing in figure 1 PCR between lacrimal gland and conjuctiva. 

Answer: We thank the reviewer for this comment, we apologize for this mistake.